# Extraction of Spatiotemporal Distribution Characteristics and Spatiotemporal Reconstruction of Rainfall Data by PCA Algorithm

Yuanyuan Liu [1,2,3,4], Yesen Liu [1,2,3,4,*], Shu Liu [1,3], Hancheng Ren [1,3], Peinan Tian [1,3] and Nana Yang [1,3]

1   China Institute of Water Resources and Hydropower Research, Beijing 100038, China; liuyy@iwhr.com (Y.L.); fcds2001@iwhr.com (S.L.); rhc_iwhr@163.com (H.R.); tianpeinan@edu.iwhr.com (P.T.); yangnana@edu.iwhr.com (N.Y.)
2   Key Laboratory of River Basin Digital Twinning of Ministry of Water Resources, Beijing 100038, China
3   Flood Control, Drought Relief and Disaster Reduction Engineering Technology Research Center of the Ministry of Water Resources, Beijing 100038, China
4   Key Laboratory of Water Safety for Beijing-Tianjin-Hebei Region of Ministry of Water Resources, Beijing 100038, China
*   Correspondence: liuys@iwhr.com; Tel.: +86-10-68781499

**Abstract:** Scientific analyses of urban flood risks are essential for evaluating urban flood insurance and designing drainage projects. Although the current rainfall monitoring system in China has a dense station network and high-precision rainfall data, the time series is short. In contrast, historical rainfall data have a longer sample time series but lower precision. This study introduced a PCA algorithm to reconstruct historical rainfall data. Based on the temporal and spatial characteristics of rainfall extracted from high-resolution rainfall data over the past decade, historical (6 h intervals) rainfall spatial data were reconstructed into high-resolution (1 h intervals) spatial data to satisfy the requirements of the urban flood risk analysis. The results showed that the average error between the reconstructed data and measured values in the high-value area was within 15% and in the low-value area was within 20%, representing decreases of approximately 65% and 40%, respectively, compared to traditional interpolation data. The reconstructed historical spatial rainfall data conformed to the temporal and spatial distribution characteristics of rainfall, improved the granularity of rainfall spatial data, and enabled the effective and reasonable extraction and summary of the fine temporal and spatial distribution characteristics of rainfall.

**Keywords:** machine learning; PCA algorithm; spatiotemporal distribution of heavy rain; feature extraction; low-resolution reconstruction; Luzhou





## 1. Introduction

In recent years, the frequency of heavy rains has increased because of global climate change, resulting in an increase in flood disasters in cities [1]. Heavy rain directly affects the distribution of floods on the underlying surface, and the intensity, duration, and spatial distribution of heavy rain are closely related to flood disasters [2,3]. The direction of the rainfall centre affects the shape of the flooding process and changes the flood peak flow. Under the same conditions of average rainfall and intensity, the peak rainfall in the middle or rear part is over 30% higher than that of uniform rainfall [4,5]. Therefore, extracting and summarising the fine temporal and spatial characteristics of rainfall and understanding the temporal and spatial variation laws of heavy rain are crucial for planning and constructing urban flood control and drainage systems, as well as improving urban flood risk management.

Rainfall data are spatial data that include both the amount and intensity of rainfall and its spatiotemporal characteristics. The extraction of rainfall features depends on the quantity and quality of the rainfall observation data. If the time series of the observation data is

longer, the distribution is denser and the interval is shorter, and the law of summarization is reasonable and fine-grained. The precise management of urban flood risk requires fine-grained rainfall features at short intervals. However, long-term rainfall data cannot reflect spatiotemporal changes in rainfall over a short period. In recent years, China's rainfall monitoring data have been of high quality, with a dense station network and minute-by-minute or hourly accuracy. In comparison, historical rainfall data have a longer time series, but the level of detail is poor—generally at intervals of 6, 12, or 24 h, which is relatively rough for analysing urban waterlogging caused by short-term heavy rainfall. In analysing and reconstructing historical rainfall data, many scholars have achieved research results of note.

Traditionally, for the reconstruction of rainfall data, linear interpolation is generally utilized; in addition to this, many scholars have utilized a variety of algorithms in the interpolation of rainfall data. Koutsoyiannis [6] discussed a variety of difference algorithms. Bijoychandra [7] generated novel 15 min precipitation datasets from hourly precipitation datasets obtained from five NA-CORDEX downscaled climate models. However, these algorithms are based on the time series of a single station and do not characterize the spatial distribution of rainstorms in the region.

Machine learning (ML)—as a branch of the AI field—has also led to revolutionary advances in many application areas [8,9], being increasingly applied in various fields over the past decade [10] that are used to identify disaster risks, manage urban floods, and address other challenges [11,12]. AI technology is also widely used in water resource management, water supply, and water diversion [13], and has played an important role in identifying flood evolution and road water accumulation [14,15]. Liu [16] estimated the precipitation induced by tropical cyclones based on machine-learning and the enhanced analogue identification of numerical prediction. In hydrology, the combination of machine-learning technology and hydrological models allows for the quantitative identification of the characteristics of basin water and sediment evolution through the coupling of basin water and sediment collection simulation technology [17].

It also has a wide range of applications in image reconstruction and enhancement; Li [18] proposed a deep learning–enhanced T1 mapping method with spatial-temporal and physical constraints (DAINTY) that can generate more accurate T1 maps and higher-quality T1-weighted images compared with other methods. Andrea [19] used a novel approach based on the U-Net deep neural network for image segmentation for the real-time extraction of tracklets from optical acquisitions. Ravishankar [20] introduced two of the most recent trends in medical image reconstruction: methods based on sparsity or low-rank models and data-driven methods based on machine learning. However, these image data, reconstructed based on machine learning techniques, do not have temporal and spatial properties.

PCA algorithms are very practical data processing algorithms in the field of machine learning [21], and have been successfully applied for feature extraction, classification, and other aspects [22]. In this study, a PCA algorithm is introduced to reconstruct historical long-term rainfall data, using Luzhou City as an example. Fine-grained features of high-resolution rainfall spatial data were extracted through dimensionality reduction, classification, and feature extraction. The historical low-resolution (6 h interval) rainfall spatial data were reconstructed into high-resolution (1 h interval) spatial rainfall data. The results show that the average error between the high value area of the reconstructed data and the measured value was within 15%, and the low value area was within 20%; the error in the high value area of the data was reduced by 45–85%, while the error in the low value area was reduced by about 10–40%, compared to traditional methods.

## 2. Materials and Methods

### 2.1. Technical Processes

In this paper, short-interval (1 h interval) rainfall data from 2013 to 2022 were used as the study samples to construct the rainfall dynamics characteristic matrix, which was

analysed and mined using a PCA algorithm to extract the refined features. The rainfall dynamic feature matrix was constructed, and the PCA algorithm was used to analyse and mine it to extract the refined features of the rainfall spatial data. Based on this feature, the long-interval (6 h interval) rainfall data from 1998 to 2012 were reconstructed to realize the feature enhancement and capacity expansion of the short-interval (1 h interval) rainfall data, which were in line with the characteristics of the rainfall spatial and temporal distribution. Based on this feature, the long-interval (6 h interval) rainfall data from 1998 to 2012 were reconstructed spatially and temporally to realize the feature enhancement and capacity expansion of short-interval (1 h interval) rainfall data that meet the desired rainfall spatial and temporal distribution characteristics. The specific technical flowchart is shown in Figure 1:

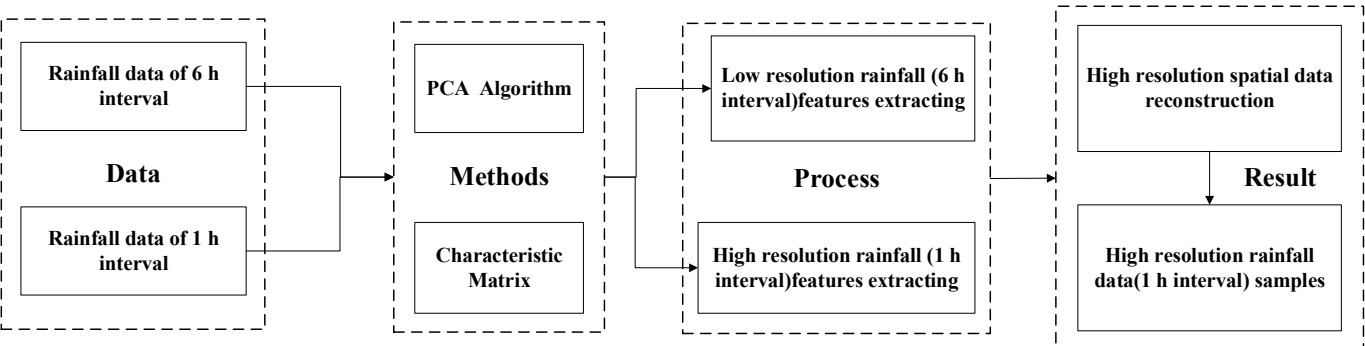

**Figure 1.** Technical Flowchart.

*2.2. Methods*

2.2.1. Constructing a Dynamic Characteristics Matrix of the Spatial and Temporal Distribution of Heavy Rainfall

To better describe the spatial and temporal distribution characteristics of heavy rainfall, this study used the proportion of rainfall at each station during each rainfall event. A dynamic characteristic matrix of the spatial and temporal distributions of heavy rainfall was constructed. Rainfall proportion is defined as the percentage of rainfall at each station during a certain period relative to the total rainfall at all stations within the research scope. The rainfall proportion was calculated using Equation (1):

$$x_{it}^j = R_{it}^j / \sum_{i=1}^s R_{it}^j \tag{1}$$

where $x_{it}^j$ is the proportion of rainfall at the $i$th rainfall station at time $t$ during the $j$th rainfall event; $R_{it}^j$ is the rainfall amount at the $i$th rainfall station at time $t$ during the $j$th rainfall event; $i = 1, 2, 3, \ldots s$, $t = 1, 2, 3, \ldots m$, $s$ is the number of rainfall stations; and $m$ is the number of time periods.

For each rainfall event, this study described the distribution characteristics of the rainfall during a certain period using a rainfall proportion matrix. Based on this method, a sample set of rainfall events was established to obtain a mathematical description of the dynamic development characteristics of multiple rainfall events in space and time, as shown in Equations (2) and (3):

$$\Omega = \{X_1, X_2, \ldots X_N\} \tag{2}$$

$$X_j = \begin{bmatrix} x_{11}^j & x_{21}^j & \cdots & x_{s1}^j \\ x_{12}^j & x_{22}^j & \cdots & x_{s2}^j \\ \vdots & & H_{it}^j & \vdots \\ x_{1m}^j & x_{2m}^j & \cdots & x_{sm}^j \end{bmatrix} \tag{3}$$

where $\Omega$ is the historical sample set of heavy rainfall events, $N$ is the number of rainfall events, $X_j$ is the proportion matrix of the $j$th rainfall event, and $x_{sm}^j$ is the proportion of rainfall at the $s$th rainfall station and the $m$th period during the $j$th rainfall event.

2.2.2. Dimensionality Reduction and Feature Extraction

Dimensionality reduction is used to reduce the dimensions or size of a matrix for a more efficient calculation. Dimensionality reduction can maintain the representation of raw data using "effective" feature data without losing the amount of intrinsic information contained in the original data [23]. In this study, a two-dimensional principal component analysis method was used for dimensionality reduction and feature extraction during rainstorms. Compared with the more commonly used one-dimensional principal component analysis, this method shows better performance in feature extraction by saving computing resources and achieving an improved recognition rate. In the calculation, it was assumed that all rainstorm events were in a low-dimensional linear space and that different events were separable in this space. Dimensionality reduction was performed in both the row and column directions of the rainstorm sample space, which means that the sample was projected in the direction with the largest variance in space; thus, spatial feature extraction and compression were realised. The calculation steps were as follows:

It was assumed that $X$ denotes the column vector matrix and that the rainfall matrix $Q$ of size $m \times n$ is directly projected onto $Y$ by a linear change, as follows:

$$Y = QX \tag{4}$$

where $Y$ is the eigenvector of the matrix $Q$ and the optimal projection axis $X$ can be determined according to the dispersion distribution of the characteristic phasor $Y$. An ideal projection matrix $X$ should ensure that the results after projection are separated as much as possible; that is, the divergence is maximised to ensure that the projection results retain the maximum amount of information. Therefore, the following criteria were used as the objective function to measure the performance of the projection matrix $X$:

$$J(X) = tr(S_x) \tag{5}$$

where $S_x$ is the covariance matrix of the training sample projection feature vector $Y$ and $tr(S_x)$ is the divergence of $S_x$. Matrix $S_x$ is defined as follows:

$$\begin{aligned}
S_x &= E\left[(Y - E(Y))(Y - E(Y))^T\right] \\
&= \mathrm{E}\left[(QX - \mathrm{E}(QX))(QX - \mathrm{E}(QX))^T\right] \\
&= \mathrm{E}[((Q - \mathrm{E}(Q))X)((Q - \mathrm{E}(Q))X)^T]
\end{aligned} \tag{6}$$

Therefore,

$$tr(Sx) = tr\left(X^T E[(Q - E(Q))^T (Q - E(Q))] X\right) \tag{7}$$

Equation (6) $E[(Q - E(Q))^T (Q - E(Q))]$ represents the covariance matrix of the sample matrix $Q$, so we defined it separately as $G_t$. If there are $N$ training sample matrices, $G_t$ can be expressed as:

$$G_t = \frac{1}{N}\sum_{i=1}^{N} \left(Q_i - \overline{Q}\right)^T \left(Q_i - \overline{Q}\right) \tag{8}$$

where $Q$ is the i-th training sample matrix, $N$ is the number of training samples, and $\overline{Q}$ is the average of all training samples, $G_t \in R^{n \times n}$. The eigenvectors of $G_t$ were extracted after Gt was formed. The eigenvectors corresponding to the cumulative contribution rate $\alpha = 0.9 \sim 0.99$ were selected to form a projection matrix $U = [u_1, u_2 \cdots u_k] \in R^{n \times k}$. Then, $F_i = Q_i \cdot U \in R^{m \times k}$, which is the projection of the sample $Q_i$ in direction U, was obtained.

That is, after feature extraction, only the number of bits in the matrix column vector was reduced, with the row vector dimension remaining unchanged.

Based on the new sample $F_i \in R^{m \times k}$, a new covariance matrix could be established:

$$G_r = \frac{1}{N} \sum_{i=1}^{N} \left( F_i - \overline{F} \right) \left( F_i - \overline{F} \right)^T \tag{9}$$

where $G_r \in R^{m \times m}$, the eigenvalues and eigenvectors of $G_r$, were calculated and the eigenvectors corresponding to the cumulative contribution rate of the eigenvalue $\alpha = 0.9 \sim 0.99$ were taken to get the projection matrix of the row direction $V = [v_1, v_2 \cdots v_k] \in R^{m \times d}$, then $V^T \cdot F_i \in R^{d \times k}$.

At this stage, the optimal projection matrices $U$ and $V$ in both directions had been obtained, the final dimension-reduced matrix of the rainstorm samples $Q_i \in R^{m \times n}$ was $Y_i = V^T \cdot Q_i \cdot U \in R^{d \times k}$, and each sample $Q_i$ has been projected onto the characteristic subspace through the optimal projection axis. Based on these treatments, the minimum distance was defined to identify the new samples.

### 2.2.3. Dynamic Clustering Analysis

The reduced sample set $Y \in R^{d \times N}$ (where $d$ is the dimension of the projected low-dimensional space and $N$ is the number of samples) was classified into $r$ subsets, where samples within each subset were similar and samples between subsets were different [24]. The features belonging to each subset were extracted by computing the centroid of each subset. This study mainly used dynamic clustering methods to classify the reduced samples. The basic idea of dynamic clustering analysis is to iteratively determine a partition scheme for $r$ clusters that minimises the total error by using the mean of these $r$ clusters to represent the corresponding samples [25,26]. In other words, this algorithm partitions the overall sample set into $r$ subsets, where samples within each subset are the most similar and samples between subsets are the most different. The mean of each subset is extracted to obtain the features belonging to that subset.

During the analysis, $r$ random sample points were selected as the initial cluster centres for $r$ subsets. The distance between all samples and these $r$ initial cluster centres was computed. The samples were then assigned to the subset whose centre was closest to them. All samples were automatically clustered into subsets based on distance, resulting in the initial classification category and initial subsets. The mean of all samples in each subset was computed to obtain a new generation of cluster centres. The distance between all samples and the new cluster centres was then computed and the samples were automatically clustered to obtain new cluster centres. The mean of all the samples in each subset was computed again and this process continued iteratively. The cluster centres and means of each subset in the $p$th and $(p+1)$th generations were compared. If the difference was within a certain range, the computation was considered to have converged, and the final subsets and cluster centres were obtained. These cluster centres were used to reconstruct low-resolution spatial data into high-resolution data based on their features.

### 2.2.4. Reconstruction of High-Resolution Spatial Data

The subsets $C = \{C_1, C_2, \cdots, C_r\}$ and their means $Z_j^{p+1}$ obtained from the above clustering method represent the feature space of the reduced data set, not the desired feature space. The PCA algorithm assumes that samples belonging to the same subset in low-dimensional space also belong to the same subset in high-dimensional space. Thus, the samples in each subset $C = \{C_1, C_2, \cdots, C_r\}$ of the low-dimensional space also belong to the same subset $B = \{B_1, B_2, \cdots, B_r\}$ in the high-dimensional space. The mean $S_j = \frac{1}{B_j} \sum_{x_i \in B_j} x_i \epsilon R^D$ of each subset in the high-dimensional space serves as the clustering centre of each class in the high-dimensional space; it represents the dynamic spatiotemporal distribution feature matrix of the samples belonging to that class.

After applying the above algorithm, the historical rainfall data were divided into several categories, and a spatiotemporal distribution feature matrix of rainfall was obtained for each category:

$$S_j = \begin{bmatrix} s_{11}^j & s_{21}^j \cdots & s_{s1}^j \\ s_{12}^j & s_{22}^j \cdots & s_{s2}^j \\ \vdots & H_{it}^j & \vdots \\ s_{1m}^j & s_{2m}^j \cdots & s_{sm}^j \end{bmatrix} \tag{10}$$

This method allows for the reconstruction of historical spatiotemporal rainfall data belonging to each class from a low resolution (6 h interval) to a high resolution (1 h interval).

### 2.3. Regional Overview

Luzhou is located in the southeastern part of Sichuan Province at the junction of the Sichuan, Chongqing, Guizhou, and Yunnan provinces, as shown in Figure 2. It is a transitional zone between the southern edge of the Sichuan Basin and the Yunnan–Guizhou Plateau. The terrain is low in the north and high in the south, with numerous rivers. The Yangtze River runs from west to east, while the Tuojiang, Yongning, Chishui, and Laixi Rivers are interlaced into a network.

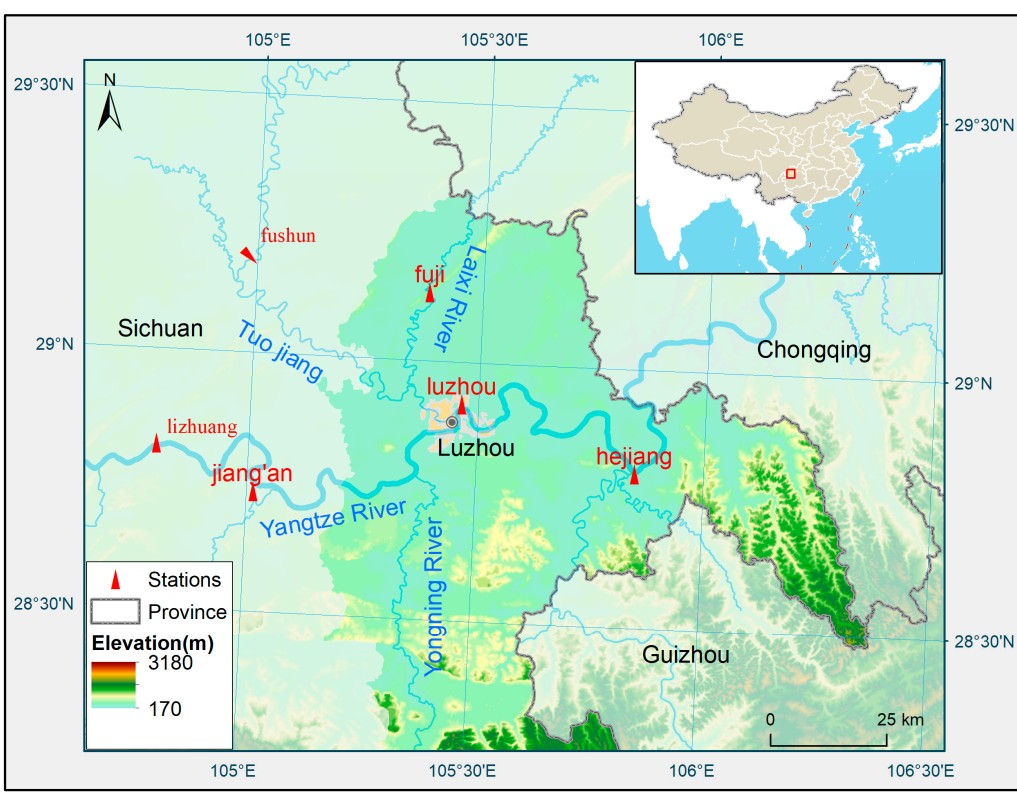

**Figure 2.** Research scope.

The region has sufficient rainfall, with an average annual rainfall of 1161 mm. However, the distribution is uneven in time and space, with 70% of the rainfall occurring between May and September. Heavy rainfall generally begins in early May and ends at the end of September.

### 2.4. Data

To gain a comprehensive understanding of the spatial and temporal distribution of rainfall in Luzhou, rainfall data from various stations were selected for this study. These included Luzhou Station, Lizhuang, and Jiang'an upstream of Luzhou on the Yangtze

River. As shown in Figure 3,Hejiang downstream of Luzhou, Fushun on the Tuojiang River upstream of Luzhou, and Fujihuo on the Laixi River.

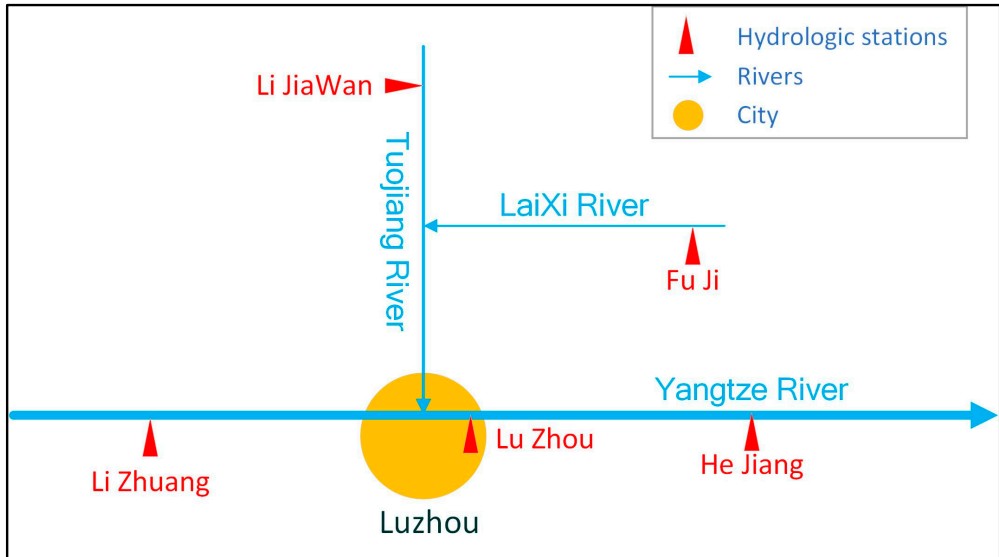

**Figure 3.** The map of hydrological stations distribution.

The data for the study in this paper are the rainfall data of the above rainfall stations from 1998 to 2022. Between 1998 and 2012, rainfall data within the region were recorded every 6 h; from 2013 to the present, they have been recorded every 1 h. Rainfall data taken every 6 h are relatively rough for analysing urban flood situations. Therefore, this study first analysed rainfall data at hourly intervals from 2013 to 2022, extracting detailed rainfall characteristics. Continuous rainfall (less than 0.1 mm for over 2 h) was considered ineffective and was used to divide rainfall events. Rainfall events with a 1 h rainfall of 30 mm or more or a 6 h rainfall of 50 mm or more were selected. Based on the above criteria and processes, 134 heavy rainfall events lasting 24 h were screened between 2013 and 2022 to form a research sample database.

## 3. Results and Discussion

This study analysed 134 heavy rain samples collected from 2013 to 2022, using the algorithms described above. The spatiotemporal distribution of heavy rainfall in Luzhou could be divided into three categories:

The first rainfall type began downstream of Luzhou City and moved upstream, reaching Luzhou City within 24 h. Due to diffusion from downstream to upstream, this type of rainfall had less impact on Luzhou City, as shown in Figure 4. The spatial and temporal characteristics of the measured rainfall process on 22 June 2003 are of this type. Figure 5 illustrates the rainfall process at 6 h intervals for this event.

The second type of rainfall occurred in the Yangtze River Basin, with the centre of heavy rain being relatively concentrated at stations such as Lizhuang and Luzhou, as shown in Figure 6. The spatial and temporal characteristics of the measured rainfall process on 11 September 2012 are of this type. Figure 7 illustrates the rainfall process at 6 h intervals for this event.

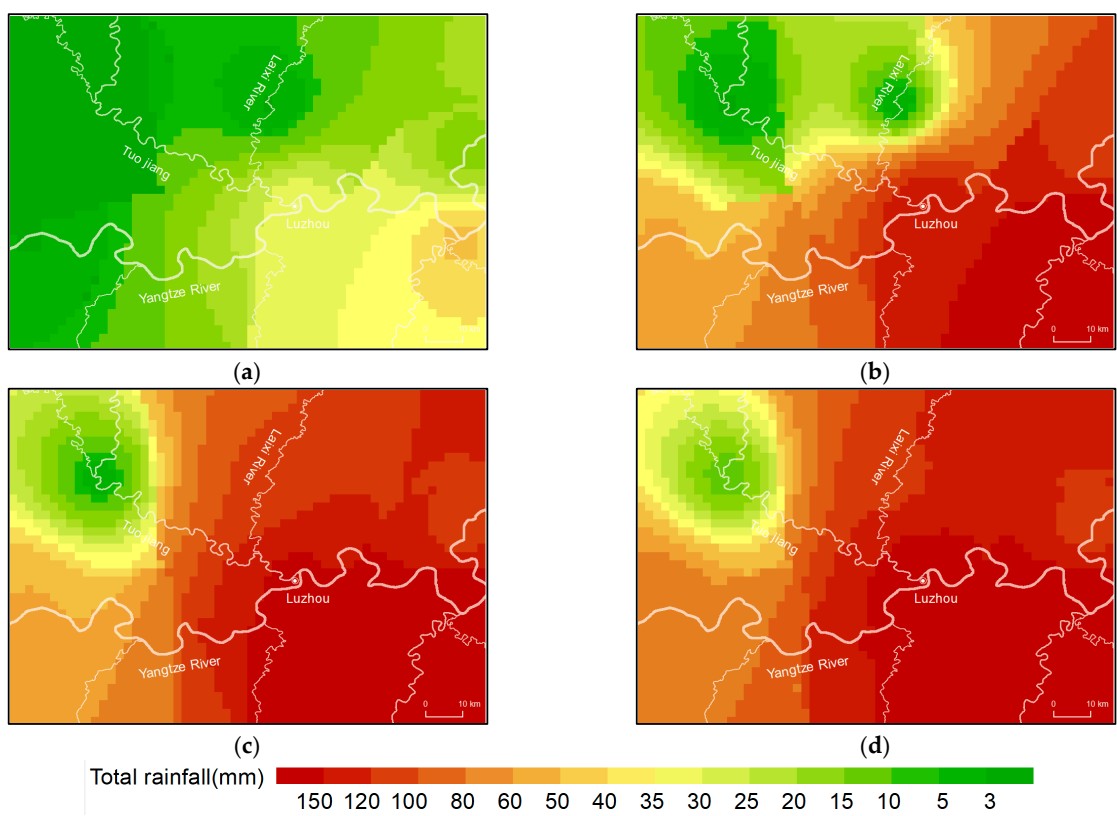

**Figure 4.** Type 1 Rainfall: (**a**) 6 h, (**b**) 12 h, (**c**) 18 h, (**d**) 24 h.

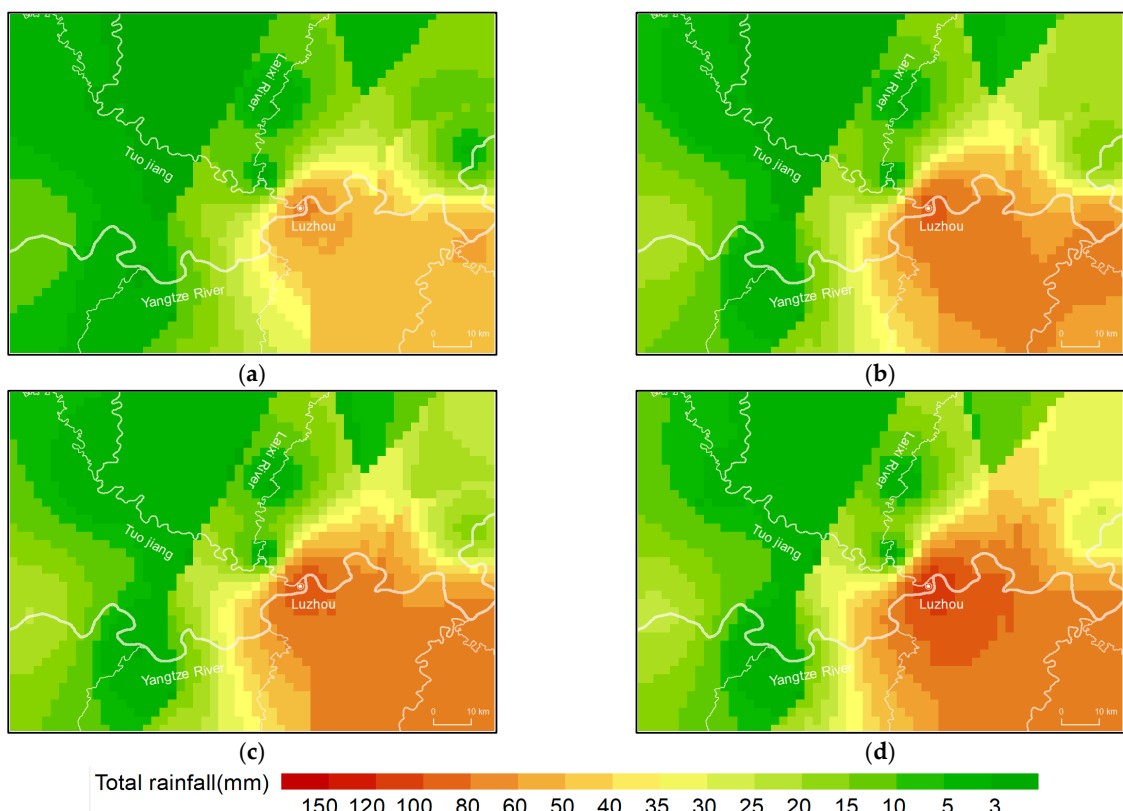

**Figure 5.** Actual Rainfall Process on 22 June 2003 (6 h rainfall): (**a**) 6 h, (**b**) 12 h, (**c**) 18 h, (**d**) 24 h.

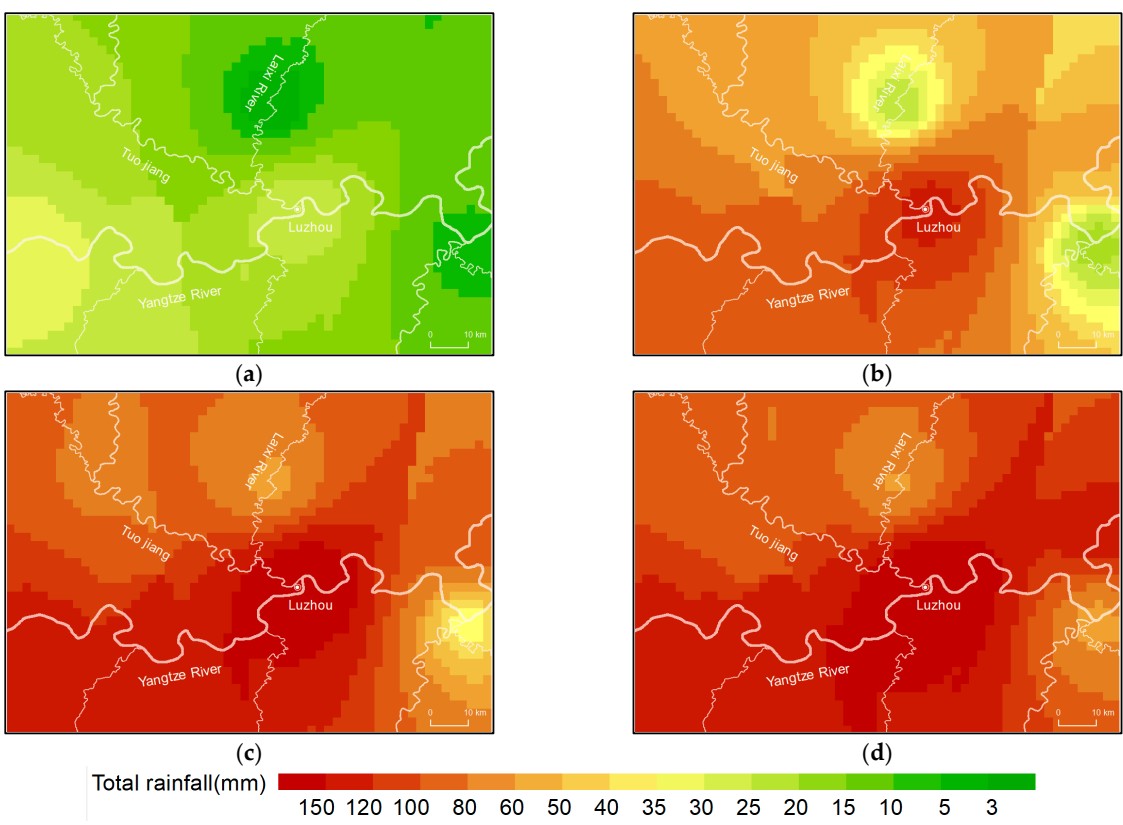

**Figure 6.** Type 2 Rainfall: (**a**) 6 h, (**b**) 12 h, (**c**) 18 h, (**d**) 24 h.

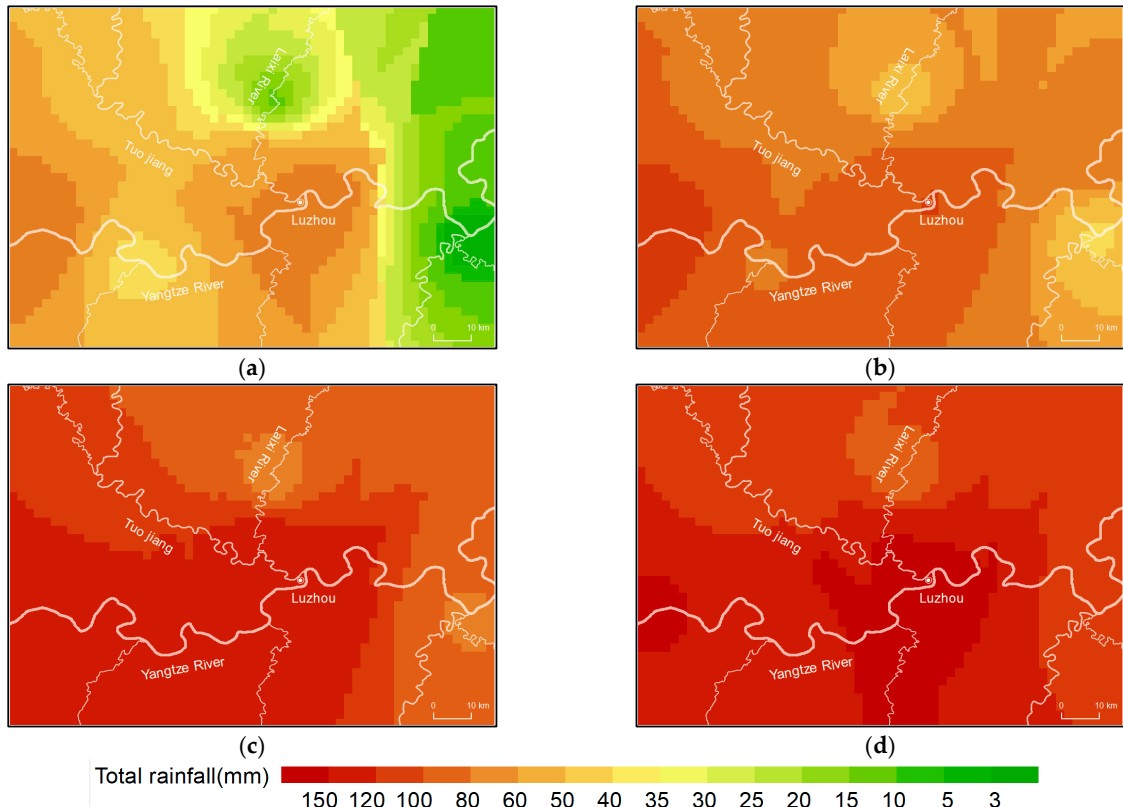

**Figure 7.** Actual Rainfall Process on 11 September 2012 (6 h rainfall): (**a**) 6 h, (**b**) 12 h, (**c**) 18 h, (**d**) 24 h.

As shown in Figure 8, the third type of rainfall had two centres of heavy rain: one upstream of the Tuojian River and the other upstream of the Yangtze River. Rainfall started upstream of the Tuojian River and combined with rainfall from upstream of the Yangtze River, moving towards Luzhou. The movement direction of the centre of heavy rain was consistent with the direction of the flood evolution of the Yangtze and Tuojian Rivers, making it the most unfavourable rainfall event. The spatial and temporal characteristics of the measured rainfall process on 12 May 2012 are of this type. Figure 9 illustrates the rainfall process at 6 h intervals for this event.

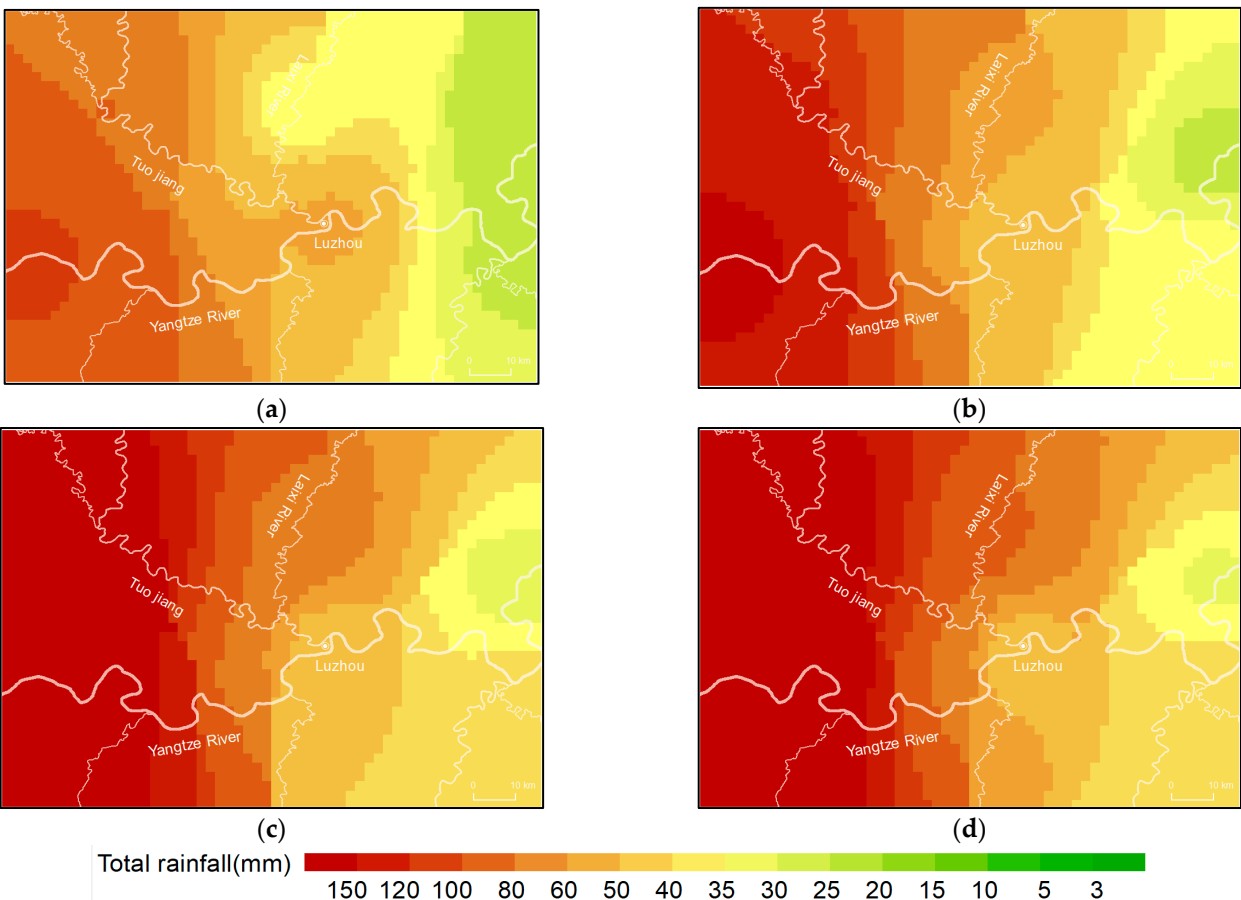

**Figure 8.** Type 3 Rainfall: (**a**) 6 h, (**b**) 12 h, (**c**) 18 h, (**d**) 24 h.

Hourly rainfall data from 2013 to 2022 were used to determine the spatiotemporal distribution characteristics of these three types of rainfall; the rainfall feature matrix was also hourly. The 6 h rainfall data from 1998 to 2012 at each station were reconstructed into hourly rainfall processes based on the extracted rainfall processes for each type. This is illustrated in Figures 10–12.

Figures 10–12 demonstrate that the method presented in this study, unlike the traditional data average interpolation encryption method, preserved the spatiotemporal distribution characteristics of rainfall while reconstructing the data.

For the first type of rainfall, during the actual rainfall process on 22 June 2003, the centre of the heavy rain appeared at the Hejiang and Luzhou stations downstream of Luzhou in the first 6 h of the rainfall event and at the Lizhuang station upstream of Luzhou at 12 h.

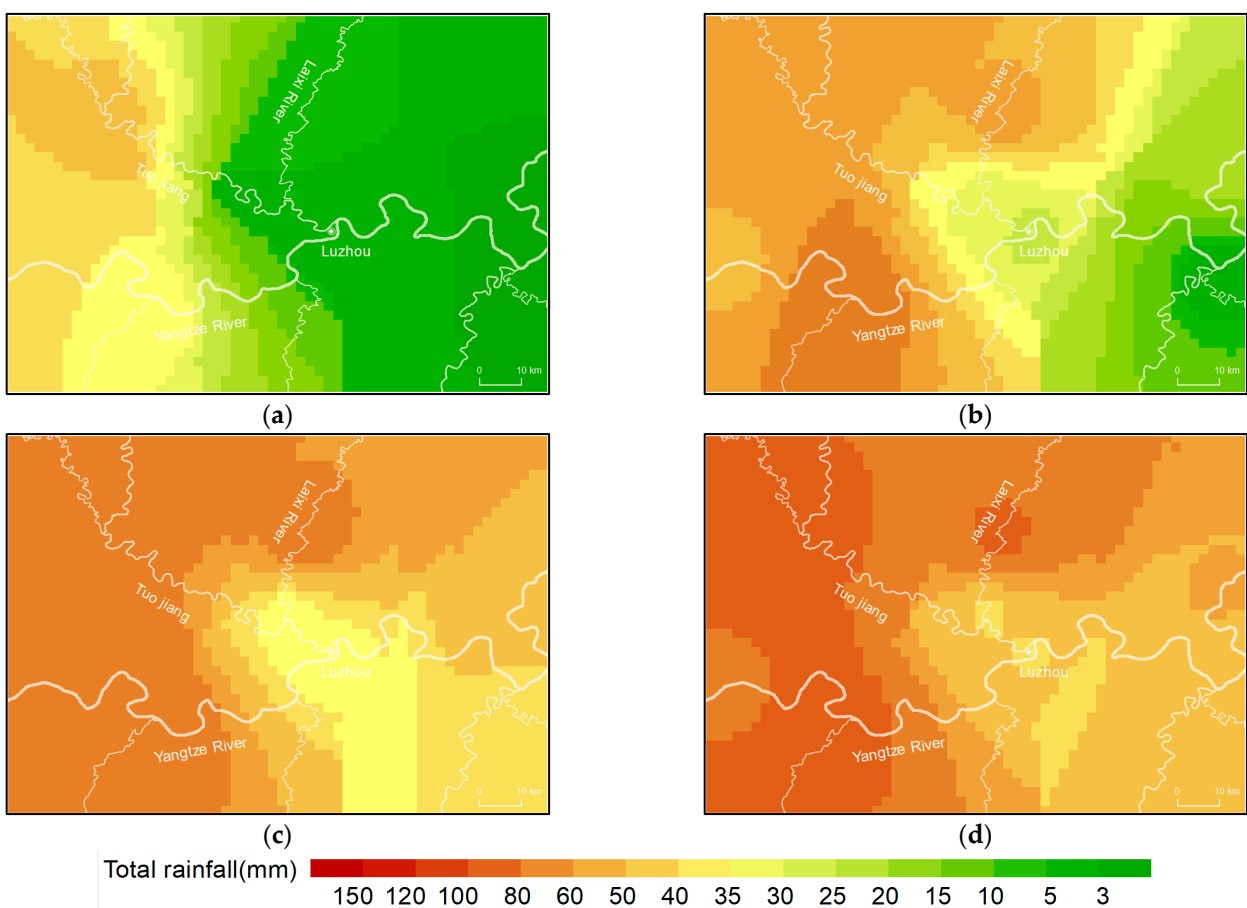

**Figure 9.** Actual Rainfall Process on 12 May 2012 (6 h rainfall): (**a**) 6 h, (**b**) 12 h, (**c**) 18 h, (**d**) 24 h.

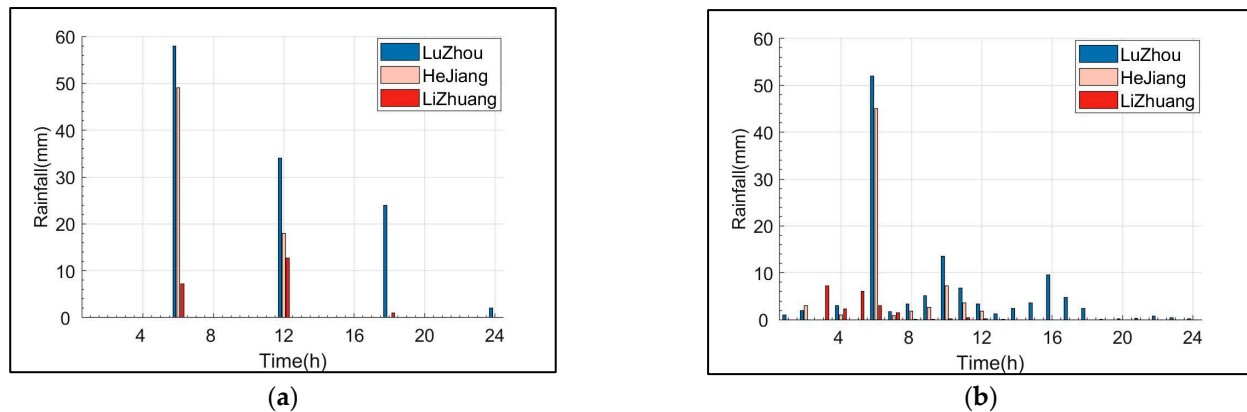

**Figure 10.** Actual Rainfall Processes at Each Station on 22 June 2003: (**a**) measured rainfall data (6 h interval) and (**b**) reconstructed rainfall data (1 h interval).

For the second type of rainfall, during the actual rainfall process on 11 September 2012, the centre of the heavy rain appeared at the Lizhuang station—upstream of Luzhou—during the first 6 h of rainfall and at the Luzhou and Hejiang stations downstream at 12 h.

For the third type of rainfall, within the first 6 h of rainfall during the actual rainfall process on 12 May 2012, the concentrated rainfall processes occurred at the Li Zhuang station upstream of Luzhou and the Fushun station upstream of the Tuojiang River. Within 18 h of rainfall, the centre of the heavy rain moved to Luzhou. These observations matched the spatiotemporal characteristics of each type of rainfall.

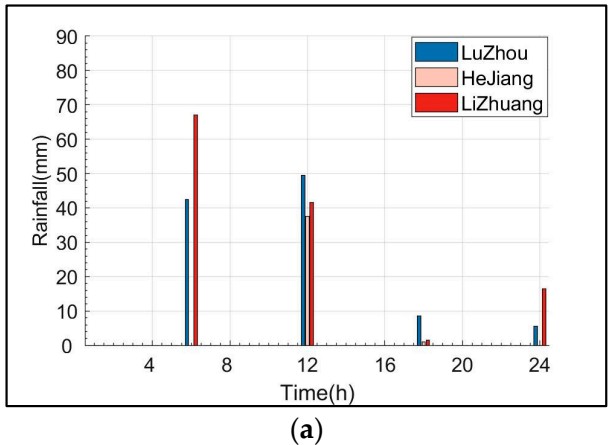
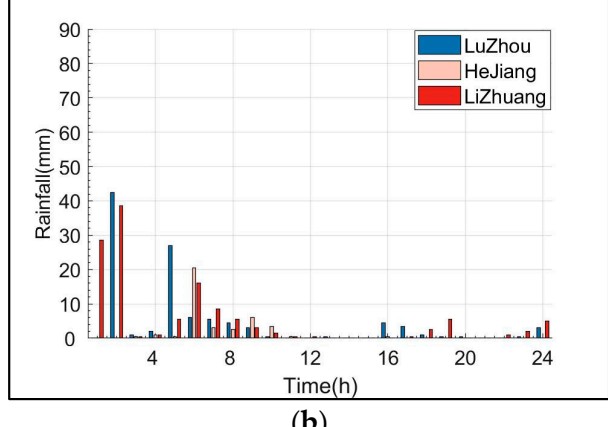

**Figure 11.** Actual Rainfall Processes at Each Station on 11 September 2012: (**a**) measured rainfall data (6 h interval) and (**b**) reconstructed rainfall data (1 h interval).

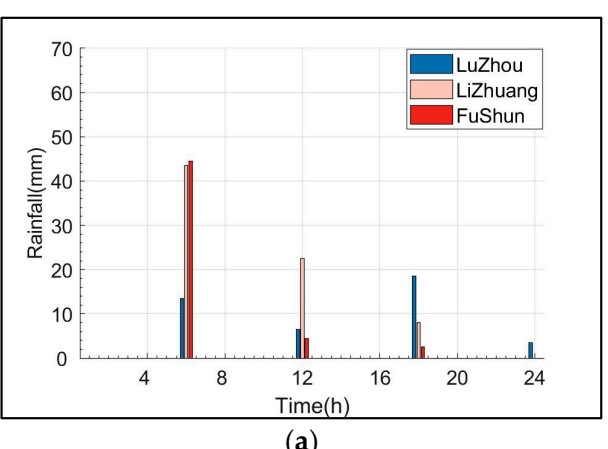
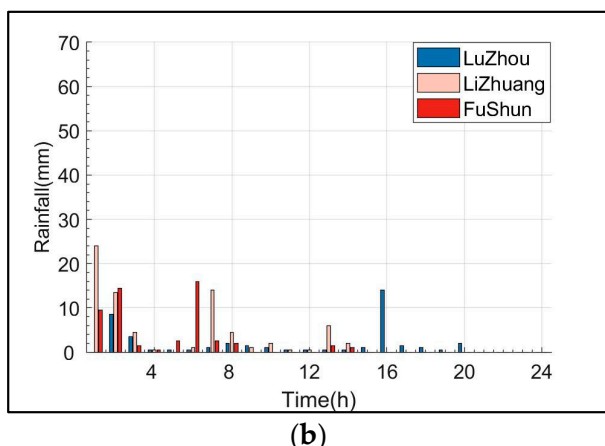

**Figure 12.** Actual Rainfall Processes at Each Station on 12 May 2012: (**a**) measured rainfall data (6 h interval) and (**b**) reconstructed rainfall data (1 h interval).

This study further validated the rationality of the method by selecting rainfall processes using the tested hourly rainfall data. During validation, the tested hourly rainfall data were first merged into 6-h periods and then reconstructed into hourly periods using the method described in this article.

The rainfall on 28 April 2013 was used as an example; this rainfall belonged to the first type of rainfall, in which the centre of the rainstorm moved downstream and upstream. Concentrated rainfall first appeared in Lizhuang, Luzhou and Hejiang, as shown in Figure 13.

As shown in Figure 13, the reconstructed data obtained using the method proposed in this study were closer to the measured data than the data processed using traditional average interpolation. The reconstructed rainfall data were more consistent with the measured values in high-value areas than in low-value areas. This study further reconstructed 24 rainfall processes over the past 10 years. A comparison of the RMSE between the reconstructed data and measured data is shown in Figure 14.

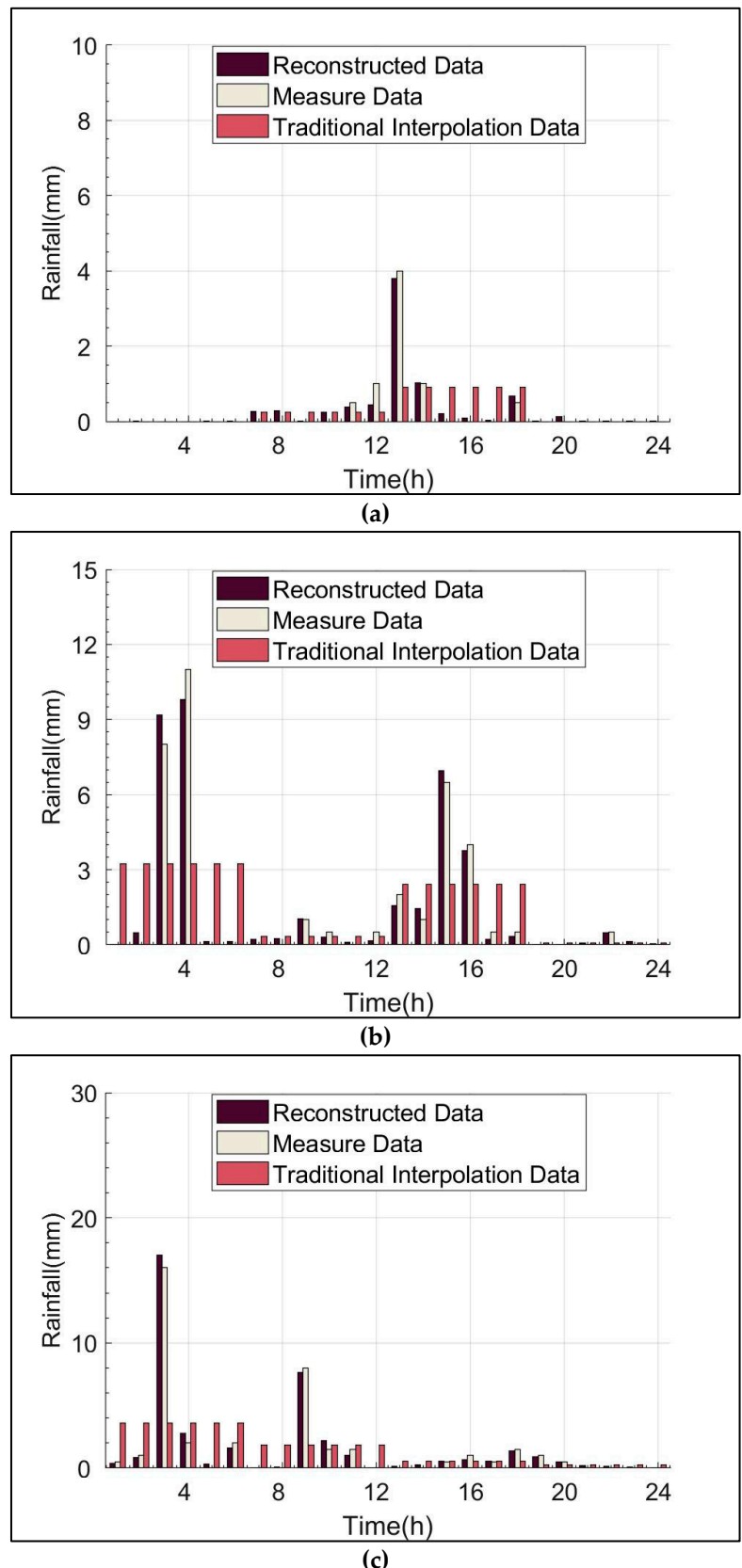

**Figure 13.** Comparison of reconstructed data, traditional interpolation results and measured data: (**a**) Lizuhang, (**b**) Luzhou, and (**c**) Hejiang.

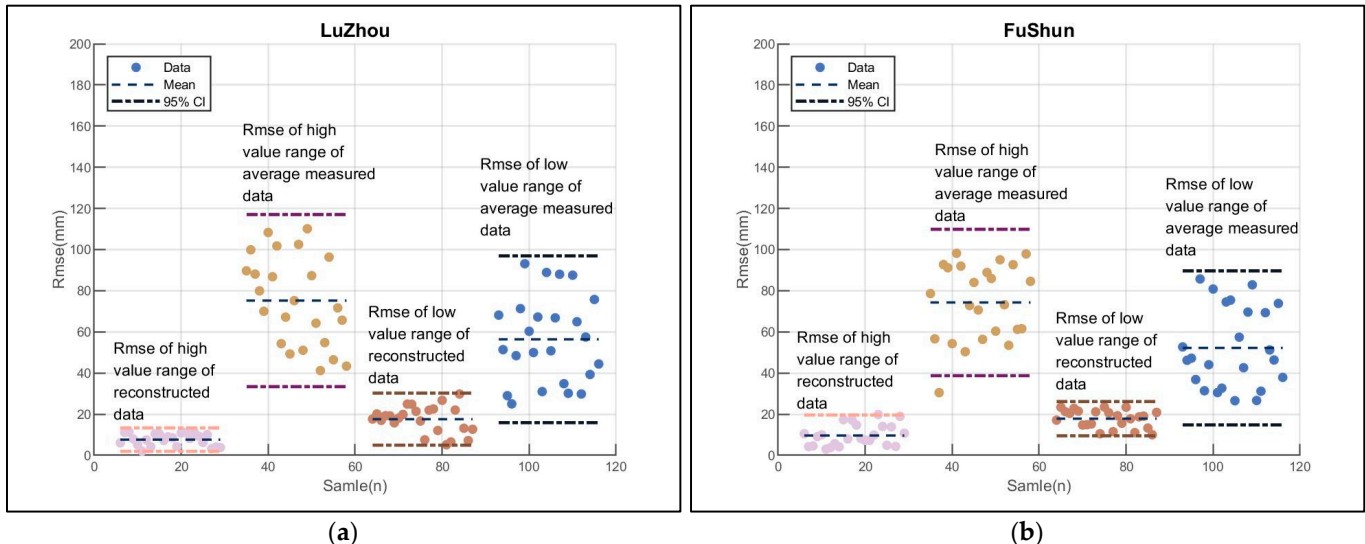

**Figure 14.** Comparison of errors between various data and measured data: (**a**) Luzhou and (**b**) Fushun.

As shown in Figure 14, the average error between the reconstructed data in the high-value area and the measured data had a 95% confidence interval of approximately 15%, and that in the low-value area was 20%. The average error between the average interpolation data in the high-value area and the measured data was approximately 60–100% with a 95% confidence interval, and that in the low-value area was approximately 30–60%. This indicates that the reconstructed data better characterised the concentrated rainfall process and spatiotemporal distribution characteristics of the rainfall. The average interpolation of the rainfall data blurred the central change process of heavy rain in the rainfall process and could not reflect the spatiotemporal distribution characteristics of the rainfall.

To summarize, compared to traditional methods of data reconstruction, the method of reconstructing rainfall data based on machine learning can reconstruct historical rainfall data into refined, hour-by-hour rainfall data. Furthermore, the reconstructed rainfall data reflects the spatial and temporal distribution characteristics of rainfall. Compared with traditional methods, the reconstructed data obtained using the method in this paper were closer to the measured values than the data processed by traditional mean interpolation, and matched the measured values better in the high-value area than in the low-value area.

## 4. Conclusions

Refined rainfall information is important for urban flood planning and management. Rebuilding low-resolution spatial data to high accuracy has significant uncertainty. Furthermore, rainfall is not only characterized by time, but also by spatial distribution. Refined rainfall data need to be able to reflect the spatial and temporal characteristics of rainfall.

This article introduced a PCA algorithm into machine learning technology for the reconstruction of low-resolution rainfall spatial data, quantitatively describing and extracting the dynamic spatiotemporal distribution characteristics of various types of rainfall, and reconstructing historical low-resolution spatial rainfall data into high-resolution rainfall spatial data. The results indicate that:

1. The PCA algorithm was successfully applied for data reconstruction with spatio-temporal attributes. Reconstruction from low-dimensional to high-dimensional data was consistent with spatiotemporal variation characteristics. The reconstructed data better reflected the concentrated rainfall process and spatiotemporal distribution characteristics of the rainfall.

2. Machine learning algorithms can extract clear features of the three types of rainfall that match the climatic characteristics of the region. The extracted features quantitatively

described the dynamic spatiotemporal distribution characteristics of the various types of rainfall.

3.　Compared with average interpolated data, the reconstructed data had a 45–85% reduction in error in high-value areas and a 10–40% reduction in low-value areas. The refined rainfall process of the reconstruction effectively reduced the error in high and low-value areas.

4.　Although the types of rainfall identified in this study are specific to Luzhou, the proposed method can be universally applied. This study used hourly rainfall spatial data. In the future, feature matrices could be extracted at the minute level to achieve a more precise reconstruction of historical rainfall data. Precise rainfall data can assist in managing urban flash flood risks, including dispatching flood prevention, emergency personnel, and material resources.

**Author Contributions:** Y.L. (Yesen Liu), P.T. and N.Y. collected and processed the data, Y.L. (Yuanyuan Liu), S.L. and H.R. proposed the model and analyzed the results, and Y.L. (Yuanyuan Liu) and Y.L. (Yesen Liu) wrote the manuscript. All authors have read and agreed to the published version of the manuscript.

**Funding:** This work is supported by the National Key R&D Program of China (2022YFC3090600), and the Chinese National Natural Science Foundation (No. 51739011).

**Data Availability Statement:** The participants of this study did not agree for their data to be shared publicly, so supporting data is not available.

**Conflicts of Interest:** The authors declare no conflict of interest.

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
