# Peer review of "Extraction of Spatiotemporal Distribution Characteristics and Spatiotemporal Reconstruction of Rainfall Data by PCA Algorithm"

_water, doi:10.3390/w15203596_

Round 1
Reviewer 1 Report (Previous Reviewer 3)
The authors have revised the Introduction which is a bit clearer than before.
However, I haven't seen any changes in the methodology and results section which are the main contribution of the paper. I would suggest the authors spend more time designing the experiments, doing literature review.

Author Response
Thank for your comment.
- Some changes were made in the last paragraph to showing the needs and targets assumed in the paper.
- In this paper, we make innovative use of machine learning algorithms to extract not only the temporal features but also the spatial features of rainfall. All the rainfall stations in the region are taken as the research object, which not only considers the temporal characteristics of rainfall, but also analyzes the spatial characteristics of rainfall movement. The extracted rainfall spatiotemporal distribution characteristics are more objective. The reconstructed historical rainfall data using the extracted spatiotemporal distribution features are more consistent.
- References have been added to the article for the years 2020-2023, namely references 20, 21 and 22. Changes were also made in the last paragraph to showing the needs and targets assumed in the paper.
Reviewer 2 Report (New Reviewer)
1. The contribution of this study is not made explicitly clear, relative to existing approaches, that is, what is new in this study that has not been done by previous studies - this is not achieved by simply summarizing what others have done.
2. The references cited lack articles from last year. So, add more references (2020-2023) to support the author's points of view. The last paragraph must be an outline of the complete study showing the needs and targets assumed in the paper.
3. The results of models should be evaluated based on performance evaluation criteria.
4. The quality of figures 10-14 is very bad.
5. There is no analysis of the extracted results and no discussion, even a simple analysis is used. The discussion section should be added to the paper, and it is required to provide some remarks to further discuss the proposed method, for example, what are the main advantages and limitations in comparison with existing methods?
Author Response
We response the reviewer’s comments point-by-point in the attachment.

Reviewer 3 Report (New Reviewer)
The authors addressed my comments and I have no further comments.
Author Response
thanks for the reviewer
Round 2
Reviewer 2 Report (New Reviewer)
Accept in present form.
This manuscript is a resubmission of an earlier submission. The following is a list of the peer review reports and author responses from that submission.
Round 1
Reviewer 1 Report
1- Page 6 the title of figure 3 change to
the map of hydrological stations distribution.
2- Pages 11 and 12 the quality of figures 10 to 13
should be better.
Reviewer 2 Report
1. Despite of many equations, the methodology is not well described.
2. It says "This study analysed 134 heavy rain samples collected from 2013 to 2022 using the 234 algorithms described above." (Line 234-235), But no any further description about how the methods are jused. For why the rainfall events are clustered into three catogories,
3. There is no analysis about how the PCA of 1 hour data in latest several could explain the rainfall in the historic period.
4. The so called "traditional interpolation method" in the paper actuall assumes the uniform distribution of hourly precipitation, which is not really traditionally used in rainfall disaggregation. In fact there are some traditional ways to disaggrate rainfall, such as the methods descirbed in many papers,(e.g., Rainfall disaggregation methods: Theory and applications by Demetris Koutsoyiannis, In book: Proceedings, Workshop on Statistical and Mathematical Methods for Hydrological Analysis (pp.1–23); Temporal disaggregation of hourly precipitation under changing climate over the Southeast United States, by Takhellambam et al. Scientific Data 2022, vol.9, 211). If making comparison, the authors should compare their results with commonly used "traditional" methods.
5. There is no effective assessment about the results in the study.
no
Reviewer 3 Report
The authors reconstructed hourly rainfall data for the Luzhou region in China by using the PCA algorithm. They claimed that their results are better than "traditional" interpolation algorithms by 65% and 40%, respectively.
I think only included the average interpolation for comparison is not sufficient. In order to prove the effectiveness of the PCA method, more interpolation methods should be included in the comparisons.
Furthermore, the result does not have the resulting spatial maps of the interpolated rainfall.
More detail comments are in the attached file.

The current manuscript is hard to read because of typo and grammar errors. Please see more details in the attached file.